# All-magnonic repeater based on bistability

Qi Wang [1] ✉, Roman Verba [2], Kristýna Davídková[3], Björn Heinz [4], Shixian Tian[5], Yiheng Rao[5,6], Mengying Guo[1], Xueyu Guo[1], Carsten Dubs [7], Philipp Pirro [4] & Andrii V. Chumak [3] ✉

Bistability, a universal phenomenon found in diverse fields such as biology, chemistry, and physics, describes a scenario in which a system has two stable equilibrium states and resets to one of the two states. The ability to switch between these two states is the basis for a wide range of applications, particularly in memory and logic operations. Here, we present a universal approach to achieve bistable switching in magnonics, the field processing data using spin waves. A pronounced bistable window is observed in a 1 μm wide magnonic conduit under an external rf drive. The system is characterized by two magnonic stable states defined as low and high spin-wave amplitude states. The switching between these two states is realized by another propagating spin wave sent into the rf driven region. This magnonic bistable switching is used to design a magnonic repeater, which receives the original decayed and distorted spin wave and regenerates a new spin wave with amplified amplitude and normalized phase. Our magnonic repeater can be installed at the inputs of each magnonic logic gate to overcome the spin-wave amplitude degradation and phase distortion during previous propagation and achieve integrated magnonic circuits or magnonic neuromorphic networks.

A base station for mobile communications is one of a most common repeaters, i.e., a device that receives a signal, cleans it up, and then retransmits it at a required power level. Repeaters play a critical role in extending the range of signals, improving signal quality, and overcoming signal degradation that can occur over long distances, and are widely used in wireless, optical, and quantum communications[1–3]. Magnonics is an emerging field in which spin waves and their quantum magnons, the collective excitation of magnetic orders, are used for data transmission and processing. Recently, spin waves have attracted much attention in the field of conventional and unconventional computing[4–7] due to their nanoscale wavelengths[8–11], controllable nonlinear phenomena[12–15], energy efficiency[16] and the abundant interaction with other quasi-/particle[17–19]. Several individual magnon-based computing devices have been demonstrated including spin-wave logic gates[20], majority gates[21], magnon transistors[22–25], magnonic directional couplers[26], and neuromorphic computing elements[27]. However, an integrated magnonic circuit cascading multiple magnonic elements has not yet been realized experimentally due to the lack of the crucial repeater to overcome the degradation (decrease in amplitude) and distortion (deformed in phase) of the spin-wave signal during propagation.

Magnon bistability refers to the phenomenon that a magnonic system can exist in one of two stable magnon states under identical external conditions. This behavior was first observed in YIG discs by Weiss[28] and further studied in μm-thick YIG films[29]. In recent years, more pronounced magnon bistabilities have been observed in cavity magnonics due to the strong coupling between magnons and cavity photons, showing potential applications in memories and switches[30–32]. However, most of the previous studies have focused on uniform precession ferromagnetic resonance (FMR), i.e., non-propagating magnons, and the size of the devices is on the order of centimeters or millimeters, which does not allow nanoscale device integration.

[1]School of Physics, Huazhong University of Science and Technology, Wuhan, China. [2]Institute of Magnetism, Kyiv, Ukraine. [3]Faculty of Physics, University of Vienna, Vienna, Austria. [4]Fachbereich Physik and Landesforschungszentrum OPTIMAS, Rheinland-Pfälzische Technische Universität Kaiserslautern-Landau, Kaiserslautern, Germany. [5]School of Microelectronics, Hubei University, Wuhan, China. [6]Hubei Yangtze Memory Laboratories, Wuhan, China. [7]INNOVENT e.V., Technologieentwicklung, Jena, Germany. ✉e-mail: williamqiwang@hust.edu.cn; andrii.chumak@univie.ac.at

In this work, based on the recently discovered bistability of the deeply nonlinear forward volume propagating spin waves excitation in nanoscale waveguides[15], we demonstrate an elegant way to switch between the two magnon states using a propagating spin wave, thus realizing a magnonic repeater. The repeater is a simple 2 μm wide strip antenna placed on top of a magnonic waveguide as shown in Fig. 1 and is proposed to be installed at the input of each magnonic logic gate to receive damped and distorted spin-wave signals from the previous level logic gate, clean them up, and then regenerate new spin waves with amplified amplitude (with the gain up to 6 times) and normalized phase to connect the next level logic gate. This opens the door to cascading magnonic logic elements with amplitude information encoding, allowing the practical realization of complex Boolean circuits predicted in theory[33] as well as the realization of magnonic synapses in neuromorphic networks.

## Results

### Concept and working principle of the magnonic repeater

The main picture of Fig. 1 shows the schematic structure of the general concept of a magnonic repeater. A 1 μm wide yttrium iron garnet (YIG) waveguide is fabricated from a 44 nm thin film using a hard mask ion beam milling technique (see "Methods")[15,34]. A coplanar waveguide (CPW) antenna, named in the later "source antenna", and a 2 μm wide strip antenna, named later "pump antenna", are placed on top of the YIG waveguide with an edge-to-edge distance of about 5 μm. The bottom inset of Fig. 1 shows the scanning electron microscope image of the experimental structure. An external field of 330 mT is applied out-of-plane along the z-axis and forward volume spin waves are investigated. Microfocused Brillouin light scattering spectroscopy (μBLS) is used to measure the spin-wave intensity at different positions along the YIG waveguide.

Figure 1 shows the schematic working principle of the magnonic repeater. Microwave pulses of the same frequency $f$ are sent to the both source and pump antennas. In principle, the excitation characteristics of both the antennas demonstrate bistable windows where the magnon state - low or high amplitude—depends on the prehistory, e.g., frequency sweep or power sweep direction (Fig. 2). This bistability is a consequence of the large nonlinearity of forward spin waves in

nanoscale waveguides[15] and is of similar nature as the bistability of a nonlinear oscillator when foldover effect takes place. At the same time, the bistable window for the used strip antenna is much wider due to the narrower k-spectrum of microwave field generated by this antenna.

This difference allows one to easily select the microwave frequency $f$ to ensure that the CPW source antenna can always excite the spin waves acting as a simple spin-wave source, and, at the same time, that the frequency $f$ is in the bistable window of the pump antenna. For the ground state (thermal level) of the magnonic waveguide, the microwave power of the pump antenna cannot be pumped into the magnonic domain due to the low excitation efficiency of the 2 μm wide antenna at the selected frequency $f$ and the large frequency gap $\triangle f$ between the excitation frequency $f$ and linear FMR frequency $f_{FMR}^{lin}$. However, when the propagating spin waves excited by the source antenna reach the strip antenna, they create a new "starting state" with nonvanishing amplitude, and if this amplitude is large enough (see "criterion below"), the magnon state under the pump antenna develops into the high-amplitude state. The source spin wave thus acts as a trigger allowing the microwave energy in the pump antenna to be pumped into the magnonic domain, thus, exciting new spin waves by the magnonic bistable switching phenomenon. More interestingly, the phase of the newly excited spin waves is determined by the phase of the microwave signal at the strip antenna, while their amplitude is purely determined by the selected excitation frequency due to the self-normalized excitation[15]. Thus, the original source signal has been cleaned up by the re-emitting process, which improves the quality and amplifies the retransmitted signal, making it suitable for the next cascade of logic elements in an integrated magnonic circuit.

### Bistable window

The curve of Fig. 2 was obtained by sending a continuous microwave current with a power of 6 dBm, swept from 4.7 GHz to 8 GHz (or vice versa) with a step size of 20 MHz, to the 2 μm wide strip antenna to excite spin waves, and the focused laser spot of the μBLS is placed about 4 μm from the edge of the strip antenna to detect the excited propagating spin waves. The frequency step is small enough so that magnon state evolves from a preceding one. As a result, the up and down frequency sweep curves do not overlap and show a hysteretic

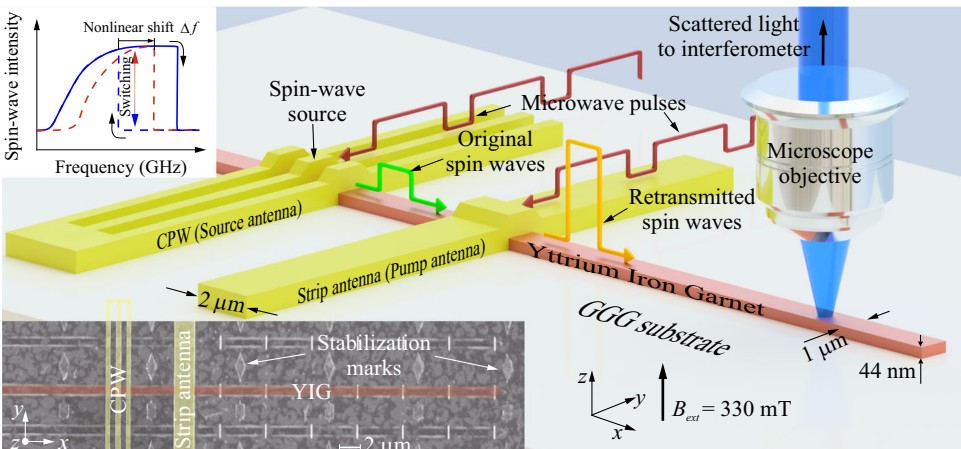

**Fig. 1 | The structure of a magnonic repeater.** Sketch of the sample and the experimental configuration: a CPW antenna (source antenna) is placed on a 1 μm wide yttrium iron garnet (YIG) waveguide to directly excite spin waves, which acts as a spin-wave source. A 2 μm wide strip antenna (pump antenna) is used to receive original spin waves and regenerate new spin waves working as a repeater. μBLS spectroscopy is employed to detect the spin-wave intensity in the YIG waveguide at different positions. The left top inset shows the schematic illustration of the bistable switching physics - source spin wave shifts the spin wave spectrum up, and, consequently, excitation characteristic of the pump antenna is shifted from the

blue dashed line to red dashed line by the nonlinear frequency shift $\Delta f$ (indicated by the black arrow) to realize the bistable switching. The left bottom inset shows the SEM image of the experimental structure. The yellow areas indicate the CPW and strip antennas. The dark red part shows the YIG waveguide. The rhombic markers are used to stabilize the sample during the measurements. The vertical nanowires on the surface of the magnonic waveguide are used to improve the detection efficiency of short wavelengths and are not used in this work. CPW coplanar waveguide. GGG gadolinium gallium garnet.

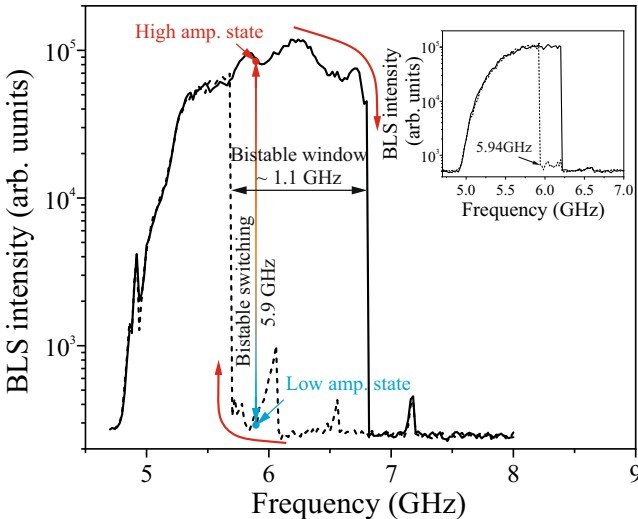

**Fig. 2 | Magnonic bistable window with two equilibrium states.** μBLS intensity as a function of excitation frequency $f$ at a microwave power of $P = 6$ dBm applied to a 2 μm wide strip antenna (pump antenna in Fig. 1) for up (solid line) and down (dashed line) frequency sweep, respectively. The frequency $f = 5.9$ GHz is used in Fig. 3 to demonstrate magnonic bistable switching. It is located in the bistable window and the corresponding two stable states are marked as high/low amplitude state. The inset shows a similar spectrum, but excited by the CPW (source) antenna. For this antenna, $f = 5.9$ GHz is located below the low-frequency edge of the bistability window and can be excited without hysteretic effects.

response resulting in a large bistable frequency window of ~1.1 GHz which is similar to our previous study[15]. The nature of this bistable window is similar to those observed in a common nonlinear oscillator (e.g., Duffing oscillator) when the foldover effect takes place, although there are certain differences (see Supplementary Materials). In addition, Fig. 2 shows several small peaks around 6 GHz, 6.6 GHz and 7.2 GHz, which belong to the higher-order spin-wave width modes that are excited by the 2 μm wide antenna with two orders of magnitude lower intensity and, therefore, can be ignored in our studies (see Supplementary Materials). The simulated snapshot of the propagating spin waves and the BLS measured profile across the waveguide clearly show that only the fundamental mode is excited in the 1 μm wide waveguide at the highest state within the bistable window (see Supplementary Materials).

**Bistable switching and magnonic repeater**

To utilize the bistability for designing a magnonic repeater, one needs an efficient and unambiguous method for the switching between the two states in the bistable window. Sure, an adiabatic frequency sweep, used for measurements of the excitation spectra Fig. 2, cannot be a method of choice. That is why we use the CPW source antenna introduced in Fig. 1 to excite propagating spin waves as a trigger to switch the bistable states under the pump antenna. The frequency of the signals applied to both the source and pump antennas was chosen to be 5.9 GHz, which is within the bistability window of the pump antenna and outside the bistability range of the source antenna (the low-frequency edge of the bistability window for the source antenna is 5.94 GHz). Thus, as discussed above, the source antenna generates spin waves always, while the pump antenna is unable to excite spin waves in the absence of a supporting source spin wave.

Figure 3 shows the general working principle of magnonic bistable switching for the realization of a magnonic repeater. First, we apply microwave pulses of this frequency with a duration of 500 ns and a repetition time of 1 μs at a power of 6 dBm to the 2 μm strip antenna (pump antenna) only, as shown in the first column of Fig. 3a. Time-resolved μBLS is used to measure the spin-wave intensity as function of

time in the frequency range from 3 GHz to 9 GHz. The laser spot is placed about 4 μm from the edge of the pump antenna as marked by blue dot. The second column of Fig. 3a shows the two-dimensional color-coded magnon spectra as a function of time, where the BLS signal (color-coded, log scale) is proportional to spin-wave intensity. Two faint horizontal lines around 4 GHz and 8 GHz are the laser side modes. No other signals indicating the excitation of spin waves are observed. A similar result is shown in the third column of Fig. 3a, where the integrated signal in the range 5–7 GHz (between the two horizontal dashed lines) is plotted. In this case, the initial state is the low-amplitude thermal background, and the system falls into the low-amplitude state, where the spin-wave amplitude is very close to the thermal level (see Fig. 2) and cannot be detected by time-resolved BLS.

Figure 3b shows the case where the same microwave pulses are sent only to the CPW antenna (source antenna). Propagating spin-wave signals are observed in the time-dependent magnon spectrum and the integrated signal–due to the high excitation efficiency at nonzero wavenumbers of the CPW antenna, it excites spin waves at 5.9 GHz independently on the prehistory, i.e., without hysteresis and bistability (see the inset of Fig. 2)[35]. Note that the source pulse arrives at the BLS measuring point with a reduced amplitude due to the spin wave damping and possible partial reflection from the metallization region formed by the pump antenna.

Figure 3c shows the case when microwave pulses were sent simultaneously to the source and pump antennas. It can be clearly seen from the time evolution of the magnon spectrum that the magnon intensity is much stronger and the thermal background has a similar noise level compared to the case of the CPW source alone, indicating that only the propagating spin waves were repeated by the pump antenna. The small tails observed at the end of each pulse are caused by the nonlinear self-phase modulation[36].

Figure 3d shows the case where the source and pump pulses have different periods to ensure that every second spin-wave pulse is reemitted by the pump antenna. Since source and pump antenna are fed with the same microwave frequency, the frequency of the spin waves after the repeater remains the same as the source frequency of 5.9 GHz. It is clear from the integrated signal that the amplification in this case is about six times. In fact, the repeater compensates for the damping of the source spin-wave signal that it has experienced during propagation to the pump antenna region. This amplification can be further improved by optimizing the excitation frequency and power. However, the main advantage of the magnonic repeater presented here is that the gain is automatically tuned by the system to maintain always the same output power of the signal after the pump region. Figure 3e shows a similar case to (d) with reserved periods. It clearly indicates that the pump antenna is only switched to the excited state when the source magnon arrives. The influences of the source and the pump powers are discussed in the Supplementary Materials, showing that the proposed method is robust with a large working window. Furthermore, the experimental results are verified by micromagnetic simulations.

The physics behind the operation of the repeater is the nonlinear frequency shift of spin waves, which is described as

$$f_k(c_k) = f_{k,0} + T_k c_k^2 \qquad (1)$$

where $f_{k,0}$ is the linear (small-amplitude) spin-wave frequency, $T_k > 0$ is the nonlinear frequency shift coefficient, and $c_k$ is canonic spin wave amplitude. In our case of forward spin waves, $c_k^2 \approx 1 - \overline{\cos\theta}$, where $\theta$ is the precession angle and overbar means averaging over the waveguide width[15]. The nonlinear frequency shift coefficient in the range from $k = 0$ to $k \approx 20$ μm$^{-1}$, which is the wavenumber of spin waves excited by the CPW antenna, weakly depends on $k$. Then, the effect of the incident spin waves can be imagined as a shift of the whole spin wave spectrum by the value $\Delta f = T_k c_k^2$[37]. The excitation characteristic of the pump

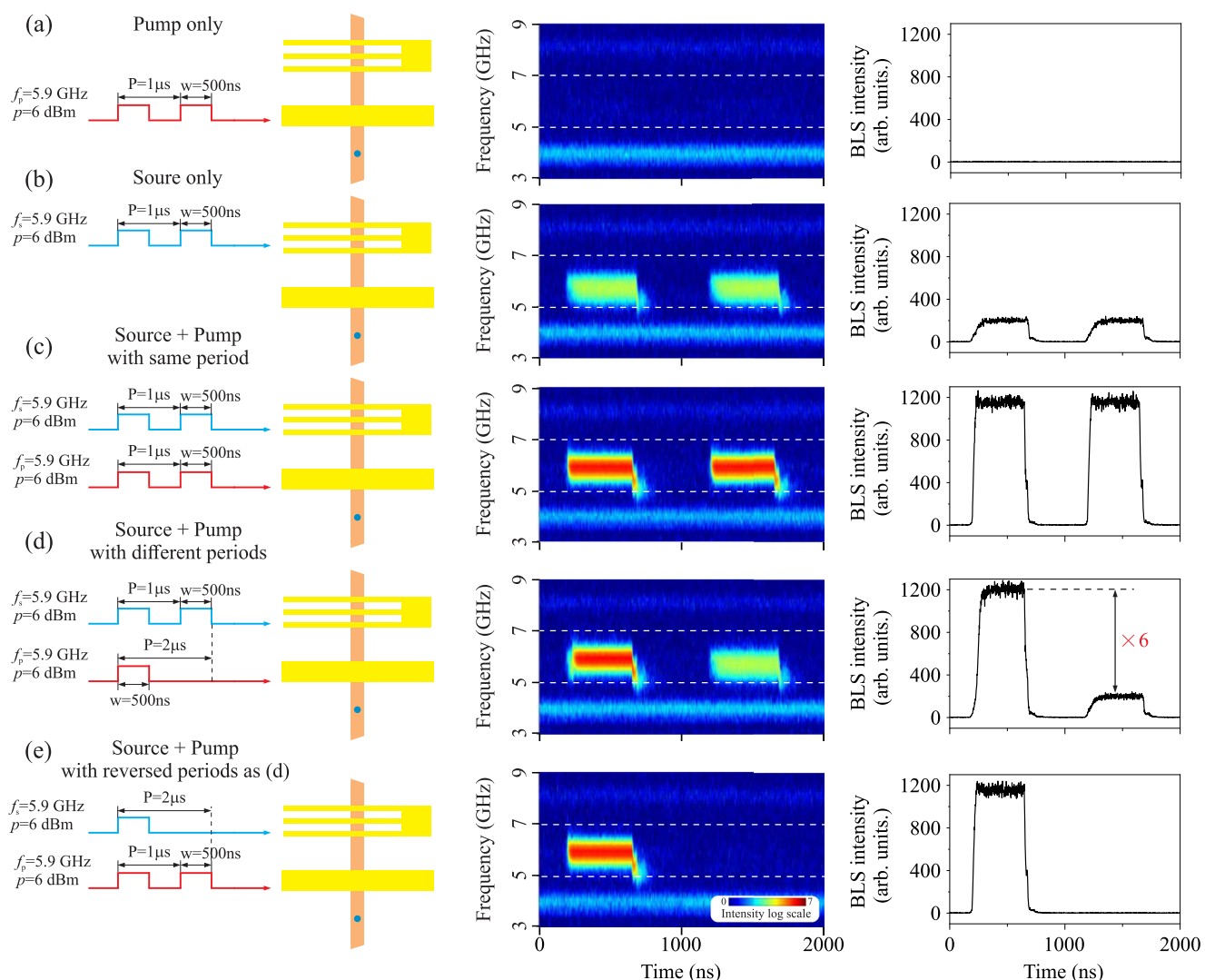

**Fig. 3 | Working principle of the magnonic repeater. a** Pump only, (**b**) Source only, (**c**) Source + pump with same period, (**d**) Source + pump with different period, (**e**) Source + pump with reversed periods as (**d**). The first column shows the schematic pictures for five different cases. The blue dots in the sample sketch indicate the (fixed) laser spot position of the BLS during the experimental measurements. The second column shows the two-dimensional BLS spectra as a function of time for two periods. The BLS signal (color-coded, log scale) is proportional to the intensity of the magnons. All spectra share the same color code. The third column shows the BLS intensity as a function of time, integrated from 5 GHz to 7 GHz, as indicated by the two horizontal dashed lines in the two-dimensional BLS spectra.

antenna (Fig. 2) is fully determined by the applied power and the shape of spin-wave dispersion curve. Thus, nonlinear shift of spin wave spectrum results (approximately) in a shift of the whole excitation characteristic by the same value $\Delta f$, as schematically shown in the inset of Fig. 1. If this shift is greater than the distance between the excitation frequency and low-frequency edge of the bistability window, the excitation frequency $f$ becomes resonant in this new characteristic (red one in the inset Fig. 1), shifted by the presence of incident spin waves. Thus, the pump antenna excites spin waves at the frequency $f$, which can be of sufficiently larger magnitude than that of the incident spin wave.

Numerical values, obtained from analytic calculations and micromagnetic simulations, are in full accordance with the above picture. In the presented case, the low-frequency edge of the bistability window is located at ~5.7 GHz, thus, 200 MHz below the excitation frequency of 5.9 GHz. The nonlinear frequency shift coefficient for $k \approx 20\ \mu m^{-1}$ is $T_k \approx 6.35$ GHz (see calculation method in ref. 15). In the simulations, a little above the switching threshold, the propagating spin waves arriving at the pump antenna have the width-averaged

precession angle of $\bar{\theta} = 12.3°$, which corresponds to canonic spin wave amplitude $c_k \approx 0.185$. Thus, incoming spin waves shift the excitation characteristic by $\Delta f \approx 220$ MHz, which is larger than the 200 MHz gap between the excitation frequency and the low-frequency edge of the bistability. As a consequence, the nonlinear shift provided by the propagating spin waves is enough to switch the pump antenna into the high amplitude excitation state.

This consideration shows that the threshold power of the incident spin waves required for switching the pump antenna into the high amplitude state is proportional to the distance between the operating frequency and the low-frequency edge of the bistability window $f_{bs}$, $c_{k,\text{th}}^2 \sim (f - f_{bs})$. It is also expected that the power and the phase of the repeated pulse is completely determined by the microwave pulse submitted to the pump antenna. Indeed, the role of the incident spin wave is just switching of the pump antenna in the bistable window from the low-amplitude state to the high-amplitude state. As soon as it has been switched, the pump antenna remains in the high-amplitude state even if the incoming spin wave pulse falls below the threshold amplitude.

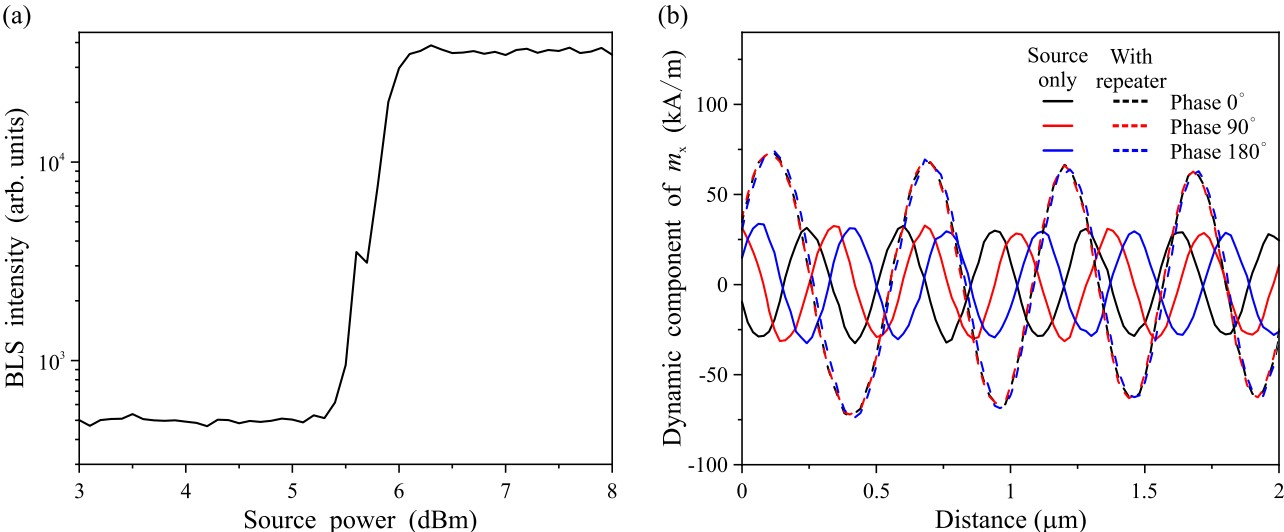

**Fig. 4 | Amplitude and phase normalization. a** The experimental results of the output spin-wave intensity as a function of the spin-wave source power. **b** The simulated spin-wave amplitude at different source phases with repeater (dashed lines) and source only (solid lines).

## Amplitude and phase normalization

In the following, we examine the amplitude and phase characteristics of the retransmitted spin waves. Figure 4a illustrates the output spin-wave intensity as a function of the source power at a pump power of 9 dBm. It is noticeable that once the source power exceeds a certain threshold (~6 dBm in our case), the output spin-wave intensity is nearly constant and is also insensitive to the pump power, as we expected from the above physical picture (see Supplementary Materials). In terms of phase information, as discussed above, the amplification is attributed to the regenerated higher amplitude spin waves by the pump antenna. Thus, the phase of the regenerated spin waves is aligned with the phase of the microwave in the pump antenna, as demonstrated in Fig. 4b. The wavelength of the spin waves is increased after the repeater due to the nonlinear upward shift of the dispersion curve[15]. The amplitude and phase normalization offers remarkable advantages, making the device into an efficient repeater to overcome the spin-wave amplitude degradation and phase distortion during its propagation. Therefore, the subsequent logic gates can directly utilize the retransmitted spin waves from the magnonic repeater as input signals without the need for further phase or amplitude modulation.

Finally, in order to prove its universality, additional micromagnetic simulations were performed with different materials (e.g., CoFeB) and varying sizes (down to 100 nm waveguide width) (see the Supplementary Materials). The consistent reproduction of the repeater behavior across different materials and sizes confirms its potential as a universal method.

## Discussion

We have experimentally observed a large bistable window of 1.1 GHz in a 1 μm-wide YIG waveguide subjected to a rf magnetic field via a strip antenna. This pump antenna provides two stable magnon states with high and low spin-wave amplitudes and allows us to switch between them. In addition, we have placed a second antenna on top of the magnonic waveguide to act as source. Spin-wave pulses emitted by the source antenna can be reemitted by the pump antenna. This is possible since the incident propagating spin waves shift the excitation characteristic of the pump antenna via the nonlinear frequency shift of spin wave spectrum. This enables to switch from the low-amplitude state in the bistable window to the high-amplitude state. This switching results in an amplification of the retransmitted spin wave as compared to the incident one, which reaches 6 times in our experiments. In addition,

the amplitude and phase of the output signal are independent of the source and pump power allowing for a simplified and robust design of magnonic circuits. Further simulations reveal that this mechanism is a universal and robust method, suitable for other materials, and can be scaled down to tens of nanometers for nanoscale circuits. Looking to the future, we would like to emphasize that the presented magnonic bistability switching can also be used for other promising applications such as neural networks and stochastic computation.

## Methods

### Nanoscale waveguide and antenna fabrication

The YIG thin film is grown on top of a 500 μm thick (111) gadolinium gallium garnet substrate by liquid phase epitaxy[38]. The parameters of the unstructured thin film were characterized by stripline vector network analyzer FMR spectroscopy and BLS spectroscopy and obtain a saturation magnetization of $M_s = (140.7 \pm 2.8)$ kA/m, Gilbert damping parameter $\alpha = (1.75 \pm 0.08) \times 10^{-4}$, inhomogeneous linewidth broadening $\mu_0 \triangle H_0 = (0.18 \pm 0.01)$ mT, and exchange constant $A_{ex} = (4.22 \pm 0.21)$ pJ/m. These parameters are typical for high quality thin YIG films[34,38]. Nanoscale YIG waveguides were fabricated using a Cr/Ti hard mask and ion beam milling process, as described in detail in refs. 15,34. The CPW antenna of a ground-signal-ground line width of 400 nm-600 nm-400 nm and an edge-to-edge spacing of 600 nm is fabricated together with a 2 μm antenna using a typical electron beam lithography technology.

### BLS measurements

A single-frequency laser with a wavelength of 457 nm is used, focused on the sample using a microscope objective (magnification 100× and numerical aperture N.A. = 0.75). The laser power of 2.8 mW is focused on the sample. A uniform out-of-plane external field of 330 mT is provided by a NdFeB permanent magnet with a diameter of 70 mm. Microwave signals with different powers and frequencies were applied to the antenna to excite and retransmit spin waves.

### Micromagnetic simulations

The micromagnetic simulations were performed by the GPU-accelerated simulation package Mumax[3], including both exchange and dipolar interactions, to calculate the space- and time-dependent magnetization dynamics in the investigated structures[39]. The parameters of a nanometer-thick YIG film were used[15,34]: saturation magnetization $M_s = 1.407 \times 10^5$ A/m, exchange constant $A = 4.2$ pJ/m. The

Gilbert damping is increased to $\alpha = 5 \times 10^{-4}$ to account for the inhomogeneous linewidth which cannot be directly plugged into Mumax$^3$ simulations. The Gilbert damping at the end of the device was set to exponentially increase to 0.5 to avoid spin-wave reflection. The mesh was set to $20 \times 20 \times 44$ nm$^3$ (single layer along the thickness) for the YIG waveguide. An external field $B_{\text{ext}} = 330$ mT is applied along the out-of-plane axis (z-axis as shown in Fig. 1) and thus sufficient to saturate the structure in this direction.

The typical parameters of CoFeB were used: saturation magnetization $M_s = 12.5 \times 10^5$ A/m, exchange constant $A = 15$ pJ/m, and the Gilbert damping $\alpha = 2 \times 10^{-3}$. The mesh was set to $5 \times 5 \times 5$ nm$^3$. An external field $B_{\text{ext}} = 2.55$ T is applied to out-of-plane.

To excite propagating spin waves, we first calculate the Oersted field distribution of a 2 μm wide strip antenna with current of 25 mA in the magneto-static approximation and plug it into Mumax$^3$ with a varying microwave frequency $f$. The $M_y(x,y,t)$ of each cell was collected over a period of 100 ns and recorded in 25 ps intervals. The fluctuations $m_y(x,y,t)$ were calculated for all cells via $m_y(x,y,t) = M_y(x,y,t) - M_y(x,y,0)$, where $M_y(x,y,0)$ corresponds to the ground state. The spin-wave dispersion curves were calculated by performing a two-dimensional fast Fourier transformation of the fluctuations.

## Data availability

All data needed to evaluate the conclusions in the paper are present in the paper and/or the Supplementary Materials. Additional data related to this paper may be requested from the authors. Source data are provided with this paper.

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

## Acknowledgements

The project is funded by the National Key Research and Development Program of China (Grant No. 2023YFA1406600), the Austrian Science Fund (FWF) via Grant No. I 4917-N (MagFunc) and Grant No. F65 (SFB PDE), the European Research Council (ERC) Proof of Concept Grant

101082020 5G-Spin and ERC Starting Grant 101042439 "CoSpiN" and the Deutsche Forschungsgemeinschaft (DFG, German Research Foundation)—271741898 and TRR 173—268565370 ("Spin + X", Project B01). Q. W. was supported by the startup grant of Huazhong University of Science and Technology Grants No. 3034012104. R.V. acknowledges support by the Ministry of Education and Science of Ukraine, project # 0124U000270 and by IEEE via "Magnetism in Ukraine Initiative" (STCU project No. 9918). Y.R. acknowledges support by the National Natural Science Foundation of China (No. 12204157) and 2023BAB139 Hubei Key R&D Program of China. We thank Ondřej Wojewoda and Michal Urbánek for the fruitful discussion on the BLS measurements.

## Author contributions

Q.W. proposed the magnonic repeater design and performed BLS measurements with help from M.G. and X.G. A.V.C. and P.P. led this project. R.V. provided theoretical support and analysis. B.H. fabricated the nanoscale YIG waveguides with help from K.D. Q.W. performed the micromagnetic simulations with help from S.T. and Y.R. C.D. grew the YIG film. Q.W. wrote the manuscript with the help of all the coauthors. All authors contributed to the scientific discussion and commented on the manuscript.

## Competing interests

The authors declare no competing interests.
