## [Peer Review File · Nature Communications]

Reviewers' Comments:

Reviewer #1:

Remarks to the Author:

The authors harnessed magnon bistability to implement a repeater and also experimentally studied its features. The structure of the proposed magnonic repeater is new, which consists of an yttrium iron garnet (YIG) waveguide, a coplanar waveguide (CPW) antenna used as the source, and a strip antenna used as the pump. These two antennas are both placed on the top of the YIG waveguide. This magnonic repeater is a direct application of the bistability of magnons in the YIG waveguide and, as mentioned by the authors, may be used as elements in integrated magnonic circuits or magnonic neuromorphic networks. The manuscript contains interesting and useful results for future applications, but it appears that the authors were not aware of previous results that are closely related to the present work. Before recommending to publish the manuscript, I suggest the authors to mention these pioneering works in the revised manuscript.

Bistability of magnon polaritons was observed in PRL 120, 057202 (2018) and PRB 174423 (2018), which is owing to the bistability of magnons. The mechanism for the magnon bistability was clearly shown to be related to the Kerr effect of magnons derived using the original microscopic model of the YIG system (see the PRL paper above and its supplemental material). Also, the transition from bistability to tristability and the memory effect were also observed in PRL 127, 183202 (2021) and, moreover, a ternary logic gate was experimentally demonstrated therein. It appears that the authors were not aware of these works, among which the first PRL paper was actually highly cited due to its innovation.

As shown in the microscopic model in PRL 120, 057202 (2018), in addition to the magnetic-field effect, the extra magnon frequency shift arises from the Kerr effect of magnons in the mean-field approximation and the bistability indicates two distinct states with different mean numbers of magnons excited in the YIG. The observed frequency shift in the present work can give an estimation of the mean magnon numbers related to the bistability.

In addition, the memory effect is an important phenomenon. Is it observed in the present work?

In Fig. 2, there are some small peaks when measuring the hysteresis loop. Is any possible mechanism for its explanation?

Reviewer #2:

Remarks to the Author:

In this paper, the authors successfully introduce an extra source antenna as a trigger to prepare the high-amplitude state of the bistable window under the pump antenna, compared to the low-amplitude state under only the pump antenna.

Previously, for the YIG waveguide only under the pump antenna, the excitation in the pump is strongly off-resonant with the linear FMR before the pump. Since the excitation pulse cannot shift the FMR nonlinearly to its excitation frequency, it's like the downward frequency sweeping and can only excite the low-amplitude state of the bistable window.

The authors cleverly introduce an extra source antenna before the pump. This source antenna can first shift the FMR nonlinearly to its excitation frequency outside its bistable window. Then the pump can incrementally shift the spin wave to the high-amplitude state of the pump's bistable window, like the gradual upward frequency sweeping with minor frequency shifts.

This paper realizes the magnonics repeater since the final generated spin wave shows increased amplitude and normalized phase, which goes one step further in the all-magnon data processing. However, as a following work of Ref. 14, some results have already been mentioned, the novelty in this paper may not be strong enough to meet the high standard of nature communications. Besides, the clarity of the paper may also need to be improved to avoid misunderstanding.

Below please find the detailed comments.

1. Fig. 2 showing the bistability of spin waves due to high-power microwave input, is already covered in Ref. 14 and provides even less information. Eq. 1 is also well explained in Ref. 14.
2. In Fig. 2, the excitation comes from either the source or pump antenna. Since the repeater is realized by excitations from both antennas, can the authors give the resulted BLS intensities under both excitations and give the connection to the repeater? Will it lead to multistability?
3. The authors may need to give the spin wave frequencies of each stage, before and after the source and pump antennas, respectively. According to Ref. 14, does the FMR correspond to 5.76 GHz under $B_0 = 330$ mT?
4. The 'switching' in this paper is the direct excitation of either the low or high amplitude

states in bistable window when the pulse is applied to only the pump antenna, or to both the source and pump antennas. This 'switching' indicates two different results at two different conditions. Since the pulse in the source antenna is applied first, there's actually no direct switching, or jump, from a former existing low amplitude state to a later high amplitude state inside the bistable window.

5. In Fig. 3, the authors may also add results for the source and pump with a reversed period as (d). This result may compare the 'switching' between low and high amplitude states of the bistable window more clearly.

6. The amplitude amplification is 6 by comparing the pump's high-amplitude state inside its bistable window and the source's state at the same excitation frequency outside its bistable window. The excitation frequency should be between the low-frequency edges of the source's and pump's bistable window from 5.7 to 5.94 GHz to set the only the pump inside the bistable window. Can the author further improve this amplification?

7. During the discussion after Eq. 1, could the author explain and clarify in the paper why the frequency shift should be greater than the gap between the excitation frequency and the low-frequency edge of the bistable window? Is the incident spin wave shifted to the excitation frequency $f = 5.9$ GHz by the source? Are the authors indicating that the source frequency can be lowered to $f_{bs} = 5.7$ GHz?

8. There exist multiple duplicate definitions of the same term, such as 'excitation frequency' and 'working frequency' and 'operating frequency', 'low-frequency edge of the bistability window' and 'left edge of the bistability window'. Please clarify these to avoid misunderstanding and improve readability.

9. In Fig. 4b, compared with the spin wave generated from the source before inputting into the pump, the spin wave after the pump increases its wavelength. Is that related to the increased precession angle (or BLS intensities)? Could the author add an explanation to the paper?

10. For Fig. 4a, if we compare it with Fig. 2B in Ref. 14, does the result only correspond to the upward power sweeping? Is there a bistable dependency on the source power?

REVIEWER COMMENTS

Reviewer #1 (Remarks to the Author):

The authors harnessed magnon bistability to implement a repeater and also experimentally studied its features. The structure of the proposed magnonic repeater is new, which consists of an yttrium iron garnet (YIG) waveguide, a coplanar waveguide (CPW) antenna used as the source, and a strip antenna used as the pump. These two antennas are both placed on the top of the YIG waveguide. This magnonic repeater is a direct application of the bistability of magnons in the YIG waveguide and, as mentioned by the authors, may be used as elements in integrated magnonic circuits or magnonic neuromorphic networks.

Thank you for the high evaluation.

The manuscript contains interesting and useful results for future applications, but it appears that the authors were not aware of previous results that are closely related to the present work. Before recommending to publish the manuscript, I suggest the authors to mention these pioneering works in the revised manuscript.

Bistability of magnon polaritons was observed in PRL 120, 057202 (2018) and PRB 174423 (2018), which is owing to the bistability of magnons. The mechanism for the magnon bistability was clearly shown to be related to the Kerr effect of magnons derived using the original microscopic model of the YIG system (see the PRL paper above and its supplemental material). Also, the transition from bistability to tristability and the memory effect were also observed in PRL 127, 183202 (2021) and, moreover, a ternary logic gate was experimentally demonstrated therein. It appears that the authors were not aware of these works, among which the first PRL paper was actually highly cited due to its innovation.

As shown in the microscopic model in PRL 120, 057202 (2018), in addition to the magnetic-field effect, the extra magnon frequency shift arises from the Kerr effect of magnons in the mean-field approximation and the bistability indicates two distinct states with different mean numbers of magnons excited in the YIG. The observed frequency shift in the present work can give an estimation of the mean magnon numbers related to the bistability.

Thank you for the suggested paper. We have added several sentences in the revised version to briefly describe the study history of magnon bistability and to discuss the difference between our presented results and this paper on page 2.

“Magnon bistability refers to the phenomenon that a magnonic system can exist in one of two stable magnon states under identical external conditions. This behaviour was first observed in YIG discs by Weiss [28] and further studied in mm-thick YIG films [29]. In recent years, more pronounced magnon bistabilities have been observed in cavity magnonics due to the strong coupling between magnons and cavity photons, showing potential applications in memories and switches [30-32]. However, most of the previous studies have focused on uniform precession ferromagnetic resonance, i.e. non-propagating magnons, and the size of the devices is on the order of centimeters or millimeters, which does not allow nanoscale device integration.”

In addition, the memory effect is an important phenomenon. Is it observed in the present work?

Yes, the switching process itself is a memory, but it is volatile, i.e., energy is required to maintain the stored information.

Figure R1 shows the working principle of a magnonic random access memory (RAM) based on bistable switching. Two microwave pulses $f_1=5.5$ GHz (write pulse) and $f_2=5.8$ GHz (bias pulse) are sent separately to the CPW antenna and stripe antenna. In these experiments, we vary the microwave power to ensure that the 5.8 GHz frequency is within the bistable window, but 5.5 GHz is outside the window and can be directly excited.

The first line shows the case where only the 5.8 GHz bias pulse is applied to the stripe antenna. No signal is detected by the BLS.

The second row shows the case where a short write pulse (20 ns duration) is applied to the CPW antenna. A small peak of comparable duration is observed in the BLS measurement.

The third line shows that both write and bias are applied separately to the CPW and stripe antennas. The stripe antenna is switched to the excited state by the short propagating spin waves. It shows that once the stripe antenna is triggered on, it starts pumping microwave energy into magnons even without a trigger pulse. The information is then stored as a form of propagating magnon to realize the all-magnon RAM similar to the all-optical RAM [Nat. Photon. **6**, 248-252(2012)].

Fig. R1 Working principle of magnonic random access memory based on bistable switching

In Fig. 2, there are some small peaks when measuring the hysteresis loop. Is any possible mechanism for its explanation?

Yes, the small peaks are the directly excited higher-order spin-wave width modes. In the waveguide, the wavelength of the spin waves is quantized along the width direction due to the finite lateral size, as can be seen in the dispersion curves below. Figure R2 shows the simulated dispersion curve in a 1 μm wide waveguide. It clearly shows several higher order width modes. The small peaks in the hysteresis loop correspond to the 7th, 9th and 11th order modes.

Fig. R2 Simulated dispersion curve with several width modes.

We discussed these on the main text page 2:

“In addition, Fig. 2 shows several small peaks around 6 GHz, 6.6 GHz and 7.2 GHz, which belong to the higher-order spin-wave width modes that are excited by the 2 μm wide antenna with two orders of magnitude lower intensity and, therefore, can be ignored in our studies.”

We also added the dispersion curve in supplementary materials and discussed it in the first section:

“Figure S1(a) shows the simulated spin-wave spectra with a foldover effect and bistable window similar to the experimental results. Several small peaks are also observed in the simulations corresponding to the higher width modes with one or two orders of magnitudes smaller intensity. Interestingly, the observed higher-order modes are the 7th, 9th and 11th modes, as confirmed by the comparison with a linear excitation spectrum at 10 times smaller driving field (see the red dashed line in Fig. S1(a)) and the simulated dispersion curve as show in Fig. S2. Simultaneously, in the nonlinear excitation regime we don't observe the excitation of 3rd and 5th modes (even modes have zero overlap with rf field and cannot be excited).”

Reviewer #2 (Remarks to the Author):

In this paper, the authors successfully introduce an extra source antenna as a trigger to prepare the high-amplitude state of the bistable window under the pump antenna, compared to the low-amplitude state under only the pump antenna.

Previously, for the YIG waveguide only under the pump antenna, the excitation in the pump is strongly off-resonant with the linear FMR before the pump. Since the excitation pulse cannot shift the FMR nonlinearly to its excitation frequency, it's like the downward frequency sweeping and can only excite the low-amplitude state of the bistable window.

The authors cleverly introduce an extra source antenna before the pump. This source antenna can first shift the FMR nonlinearly to its excitation frequency outside its bistable window. Then the pump can incrementally shift the spin wave to the high-amplitude state of the pump's bistable window, like the gradual upward frequency sweeping with minor frequency shifts.

This paper realizes the magnonics repeater since the final generated spin wave shows increased amplitude and normalized phase, which goes one step further in the all-magnon data processing.

Thank you for the high evaluations.

However, as a following work of Ref. 14, some results have already been mentioned, the novelty in this paper may not be strong enough to meet the high standard of nature communications.

We appreciate that the reviewer raising this discussion.

In Ref. 15 (in the revised version), we observed two phenomena:(1) deeply nonlinear frequency shift and (2) bistable window in nanoscale waveguide due to the large precession angle of FVSWs.

In that work, we only used the deeply nonlinear frequency shift to efficiently excite short wavelength spin waves and the bistability of the system was observed, but not necessary for the excitation of the short spin waves.

In fact, the bistable window itself, without switching between the two equilibrium states, does not guarantee any applications. An efficient switching method between the two states is even more important than the bistable window itself. In this paper, we introduce an additional source (CPW) antenna to excite propagating spin waves which are able to trigger the nonlinear state under the pump antenna.

To the best of our knowledge, this is the first time that the switching between the two states of the bistability is not realized via sweeping the driving power or the magnetic field

applied to the bistability region. And it is precisely this new phenomenon that opens up access to the application: the triggering spin wave signal is re-emitted with the same phase but increased amplitude (up to a certain fixed level) and can be sent on to the next cascade of magnonic logic elements in an integrated circuit. As mentioned by the reviewer, this is a significant step forward in all-magnon data processing.

In addition, the magnonic repeater is only one of the applications of the switching. The all-magnon bistable switching phenomenon proposed in this paper opens a door to the realization of various magnonic devices, such as magnonic random access memories as shown in the response letter to reviewer 1, magnonic random number generators (see the reply to Q2 below), and magnonic neurons, etc.

Therefore, we strongly believe that this paper is a pioneering work for the realization of all-magnonic circuits and deserves to be published in Nature Communications.

Besides, the clarity of the paper may also need to be improved to avoid misunderstanding.

Below please find the detailed comments.

1. Fig. 2 showing the bistability of spin waves due to high-power microwave input, is already covered in Ref. 14 and provides even less information. Eq. 1 is also well explained in Ref. 14.

In this paper, the width of the waveguide is increased from 200 nm (Ref. 15) to 1 μm in order to increase the transmission from the source to the pump antenna. As a result, the spin-wave spectra are different from previous studies. (Note that Figure 2 in this paper is not the repetition of the figure in Ref. 15. It is a new result obtained from a new sample with a different waveguide width.) The idea here is to present the most important information to help the reader understand the basic physics without having read the previous paper.

2. In Fig. 2, the excitation comes from either the source or pump antenna. Since the repeater is realized by excitations from both antennas, can the authors give the resulted BLS intensities under both excitations and give the connection to the repeater? Will it lead to multistability?

Figure 4(a) in the main text shows the experimental results at the output for excitations from both antennas. When the source power is large enough ($\sim 6\text{dBm}$ in this case), the output spin-wave intensity is almost constant. Interestingly, an intermediate spin-wave intensity is observed when the source power is between 5.5 dBm and 6 dBm, which should not occur in bistable switching.

The intermediate spin-wave intensity is caused by the stochastic nature of bistable switching. In order to collect enough signal, the BLS will measure approximately 30 s (3×10^7 pulses) for each point. For source powers between 5.5 dBm and 6 dBm, the pump

pulses do not trigger the high amplitude state with 100% probability. Thus, the averaged spin-wave intensity appeared between the two states.

To check this, we performed a similar experiment but increased the pulse duration drastically to 2.9 s (period 3 s) of the pump pulse but the duration of the trigger pulse is still only 50 ns (period 3 s). Once the pump antenna is triggered on, it remains on until the end of the pulse duration, similar to the third row of Fig. R1. The duration of 2.9 s time is sufficient for the BLS to collect enough data. Figure R3(a) shows the similar experimental results to Figure 4(a) but for different conditions. Figure R3(b) shows the probability of the three special source powers indicated in Fig. R3(a) using different colors. The total time is 600 s (200 repetitions). It is clear that the magnonic bistable switching is a stochastic process.

Figure R3 (a) The experimental results of the output spin-wave intensity as a function of the spin-wave source power. (b) The probability of magnonic bistable switching.

In conclusion, we have not observed multistability in this system. However, the switching process is stochastic. In this study, we have chosen the power high enough to ensure that the switching is deterministic. Stochastic switching will be studied systematically in the future.

3. The authors may need to give the spin wave frequencies of each stage, before and after the source and pump antennas, respectively. According to Ref. 14, does the FMR correspond to 5.76 GHz under $B_0 = 330$ mT?

The middle column of Figure 3(b)-(d) in the main text shows the two-dimensional BLS measurements. The x-axis is time and the y-axis is frequency. Figure 3(b) shows the case where only the source is applied to the CPW antenna. The frequency of the spin waves is around 5.9 GHz corresponding to our excitation. Figure 3(c) shows the same frequency for the case with both source and pump power. The small tails observed at the end of each pulse are caused by the nonlinear self-phase modulation.

We have added a sentence on page 6 to highlight the frequency consistency.

“Since source and pump antenna are fed with the same microwave frequency, the frequency of the spin waves after the repeater remains the same as the source frequency of 5.9 GHz.”

As mentioned above, in this paper the width of the waveguide is increased from 200 nm (Ref. 15) to 1 μm to achieve more signal transmission from the source to the pump antenna. The FMR frequency obtained from micromagnetic simulations is 4.64 GHz at $B_0=330$ mT, close to the experimental results of around 4.76 GHz (the frequency at which the BLS intensity starts to increase in Fig. 2). The reason for the lower frequency compared to Ref. 15 is the larger out-of-plane demagnetization factor due to the increased waveguide width.

4. The ‘switching’ in this paper is the direct excitation of either the low or high amplitude states in bistable window when the pulse is applied to only the pump antenna, or to both the source and pump antennas. This ‘switching’ indicates two different results at two different conditions. Since the pulse in the source antenna is applied first, there’s actually no direct switching, or jump, from a former existing low amplitude state to a later high amplitude state inside the bistable window.

In this paper, “switching” refers to the change from the low amplitude to the high amplitude state under the pump antenna. The spin waves from the source antenna only induce/trigger the switching under the pump antenna. Thus, the switching does not take place below the source antenna. As shown in Fig. 3(a-c), a microwave pulse is applied to the strip (pump) antenna and cannot directly excite spin waves. The pump antenna is only switched to the high amplitude state when the source spin wave arrives. In response to the reviewer’s next question, we have added an extra line in Fig. 3 to clearly show this switching.

5. In Fig. 3, the authors may also add results for the source and pump with a reversed period as (d). This result may compare the ‘switching’ between low and high amplitude states of the bistable window more clearly.

Thank you for the suggestion. Now, we added this information into Fig. 3(e) and corresponding description on page 6.

“Figure 3(e) shows a similar case to (d) with reserved periods. It shows that the pump antenna is only switched to the excited state when the source magnon arrives.”

6. The amplitude amplification is 6 by comparing the pump’s high-amplitude state inside its bistable window and the source’s state at the same excitation frequency outside its bistable window. The excitation frequency should be between the low-frequency edges of the source’s and pump’s bistable window

from 5.7 to 5.94 GHz to set the only the pump inside the bistable window. Can the author further improve this amplification?

In our studies, spin waves emitted from the pump antenna are excited by the deeply nonlinear shift. This spin wave intensity does not depend on the input microwave power, but on the excitation frequency. Therefore, one way to improve the amplification is to increase the excitation frequency. As the reviewer already mentioned, the frequency should meet two criteria: (1) it should be lower than the low-frequency edge of the bistable window of the source (2) it should be within the bistable window of the pump. In our case, it should be between 5.7 GHz and 5.94 GHz. However, the low-frequency edge of the bistable window of the source can be further increased by increasing the source power (it is not necessary to keep the source and pump power equal.) Even, the bistable window of the pump can be shifted up by increasing the pump power.

To increase the amplification, (1) we can increase the output amplitude of the spin wave by increase the excitation frequency, (2) decrease the threshold spin-wave amplitude to trigger the switching by optimizing the pump power (move the low-frequency edge of the bistable window of the pump antenna close to the excitation frequency.)

We have added a general sentence on page 6 to point this out.

“This amplification can be further improved by optimizing the excitation frequency and power.”

7. During the discussion after Eq. 1, could the author explain and clarify in the paper why the frequency shift should be greater than the gap between the excitation frequency and the low-frequency edge of the bistable window?

Is the incident spin wave shifted to the excitation frequency $f = 5.9$ GHz by the source?

Are the authors indicating that the source frequency can be lowered to $f_{bs} = 5.7$ GHz?

The frequencies of both the incident spin waves and the microwave applied to the pump antenna are the same equal to 5.9 GHz, as explained in the answer to question No. 3. There is no frequency conversion in our experiment.

In the revised version, we extend the explanation following Eq. (1). The main physics is that the incoming spin wave shifts the whole spin-wave spectrum up. Then the low-frequency edge of the bistability is not only defined by the pump antenna itself, but also by the intensity of the incoming spin waves. The incoming spin waves shift the whole spectrum up (both the FMR frequency and the low-frequency edge of the bistable window), and if this shift is larger than the gap between the excitation frequency and the low-frequency edge in the absence of the source wave, than in its presence the excitation frequency appears to the left of the shifted low-frequency bistability edge, i.e. in a single-state region where only the high amplitude state is present. We have also added a schematic to the inset of Fig. 1, which we believe will help to understand the idea.

Regarding the source frequency, if we only want to switch the pump antenna, the source frequency can be any frequency higher or lower than 5.9 GHz, the only requirement is that it is outside the bistability range of the source CPW antenna. Here, we want to demonstrate the application as a repeater, in which the spin wave frequency after the repeater should be the same as before. Therefore, the source frequency should be the same as the pump one. But they can be simultaneously varied within the bistability range of the pump antenna, with the additional requirement of being in single-state range of the CPW antenna. In our device, this range of choice is between 5.7 GHz and 5.94 GHz. In other applications, such as magnonic random access memory, the frequency of the source may be different compared to the pump.

8. There exist multiple duplicate definitions of the same term, such as 'excitation frequency' and 'working frequency' and 'operating frequency', 'low-frequency edge of the bistability window' and 'left edge of the bistability window'. Please clarify these to avoid misunderstanding and improve readability.

Thanks for the suggestion. We have standardized our terminology to avoid ambiguity. We now use “excitation frequency” to replace “working frequency and operating frequency” and use “low-frequency edge of the bistability window” to replace “left edge of the bistability window”.

9. In Fig. 4b, compared with the spin wave generated from the source before inputting into the pump, the spin wave after the pump increases its wavelength. Is that related to the increased precession angle (or BLS intensities)? Could the author add an explanation to the paper?

Thanks for pointing that out. In our system the spin waves are excited by a deeply nonlinear shift. The physics is shown in Fig. 3 in Ref. 15. As the intensity of the spin wave increases, the dispersion curve shifts upward. However, the frequency of the spin wave is fixed by the energy conservation. Thus, the wavenumber of the spin waves is decreased (wavelength increases) when they are amplified.

We have added the explanation on page 8.

“The wavelength of the spin waves is increased after the repeater due to the nonlinear upward shift of the dispersion curve [15].”

10. For Fig. 4a, if we compare it with Fig. 2B in Ref. 14, does the result only correspond to the upward power sweeping? Is there a bistable dependency on the source power?

The difference between the Fig. 2B (Ref. 15) and Fig. 4a are following:

(1) Fig. 2B (Ref. 15) is obtained by applying only one continue waves sweeping. In this case, a small power step is chosen so that the magnon state evolves from a previous

one. As a result, the up and down frequency sweep curves do not overlap and show a hysteretic response resulting in a bistable window.

- (2) Fig. 4a is obtained by applying two pulses, the x-axis is the power of the trigger pulse. However, there is no hysteresis loop when the applied microwave power is pulsed. At the end of each pulse, the spin waves decay and the system returns to thermal ground. Therefore, the excitation always starts from the thermal ground. Figure R4 shows the case where the upward and downward power sweeps almost overlap when a microwave pulse is used.

Figure R4 The BLS intensity as a function input microwave power for both upward and downward sweep.

Reviewers' Comments:

Reviewer #1:

Remarks to the Author:

The authors satisfactorily answered my questions and accordingly improved the manuscript. I can now recommend this revised manuscript for publication in Nature Communications.

Reviewer #2:

Remarks to the Author:

The authors have responded to my previous concerns accordingly and clarified that the novelty lies in the switching mechanism rather than bistability.

REVIEWER COMMENTS

Reviewer #1 (Remarks to the Author):

The authors satisfactorily answered my questions and accordingly improved the manuscript. I can now recommend this revised manuscript for publication in Nature Communications.

We thank this Reviewer for supporting the publication of our manuscript in Nature Communications.

Reviewer #2 (Remarks to the Author):

The authors have responded to my previous concerns accordingly and clarified that the novelty lies in the switching mechanism rather than bistability.

We thank this Reviewer for supporting the publication of our manuscript in Nature Communications.